# Ohr and OhrR Are Critical for Organic Peroxide Resistance and Symbiosis in *Azorhizobium caulinodans* ORS571

**DOI:** 10.3390/genes11030335

**Published:** 2020-03-20

**Authors:** Yang Si, Dongsen Guo, Shuoxue Deng, Xiuming Lu, Juanjuan Zhu, Bei Rao, Yajun Cao, Gaofei Jiang, Daogeng Yu, Zengtao Zhong, Jun Zhu

**Affiliations:** 1Department of Microbiology, College of Life Sciences, Nanjing Agricultural University, Nanjing 210095, China; 2018216028@njau.edu.cn (Y.S.); 2017116097@njau.edu.cn (D.G.); 2017116094@njau.edu.cn (S.D.); 2017116095@njau.edu.cn (X.L.); rabecca9003@sina.com (B.R.); caoyajun@njau.edu.cn (Y.C.); jun_zhu@njau.edu.cn (J.Z.); 2Jiangsu Provincial Key Lab for Organic Solid Waste Utilization, Jiangsu Collaborative Innovation Center for Solid Organic Waste Resource Utilization, National Engineering Research Center for Organic-based Fertilizers, Postdoctoral Station of Agricultural Resources and Environment, Nanjing Agricultural University, Nanjing 210095, China; gjiang@njau.edu.cn; 3Tropical Crops Genetic Resources Institute, Chinese Academy of Tropical Agricultural Science, Danzhou, Hainan 571737, China; geng0209@126.com

**Keywords:** *Azorhizobium caulinodans*, Ohr, OhrR, organic peroxide, symbiosis

## Abstract

*Azorhizobium caulinodans* is a symbiotic nitrogen-fixing bacterium that forms both root and stem nodules on *Sesbania rostrata*. During nodule formation, bacteria have to withstand organic peroxides that are produced by plant. Previous studies have elaborated on resistance to these oxygen radicals in several bacteria; however, to the best of our knowledge, none have investigated this process in *A. caulinodans*. In this study, we identified and characterised the organic hydroperoxide resistance gene *ohr* (AZC_2977) and its regulator *ohrR* (AZC_3555) in *A. caulinodans* ORS571. Hypersensitivity to organic hydroperoxide was observed in an *ohr* mutant. While using a *lacZ*-based reporter system, we revealed that OhrR repressed the expression of *ohr*. Moreover, electrophoretic mobility shift assays demonstrated that OhrR regulated *ohr* by direct binding to its promoter region. We showed that this binding was prevented by OhrR oxidation under aerobic conditions, which promoted OhrR dimerization and the activation of *ohr*. Furthermore, we showed that one of the two conserved cysteine residues in OhrR, Cys_11_, was critical for the sensitivity to organic hydroperoxides. Plant assays revealed that the inactivation of Ohr decreased the number of stem nodules and nitrogenase activity. Our data demonstrated that Ohr and OhrR are required for protecting *A. caulinodans* from organic hydroperoxide stress and play an important role in the interaction of the bacterium with plants. The results that were obtained in our study suggested that a thiol-based switch in *A. caulinodans* might sense host organic peroxide signals and enhance symbiosis.

## 1. Introduction

Reactive oxygen species (ROS), such as hydroperoxides, organic peroxides, and free radicals, are continuously produced as by-products of various metabolic pathways. ROS are highly reactive and toxic [1,2], which causes damage to proteins, lipids, and DNA, eventually leading to cell death [3,4]. Furthermore, ROS can affect the interaction between microorganism and host [5], particularly in microbes that are hypersensitive to ROS [6]. Rhizobia, a type of aerobic Gram-negative bacteria, are able to reduce atmospheric nitrogen through a symbiotic interaction with a host legume [7]. The symbiont is a major contributor to soil nitrogen fertility and the global nitrogen cycle [8]. During the early phases of legume-rhizobia symbiosis, plants generate ROS to defend against the invading species [9,10]. Endogenous ROS mediates the Nod factor-induced root hair formation has been observed in *S.rostrata* or a few other legumes [11]. Nod factor has also been found to suppress the activity of the ROS-generating system in a host legume [10]. In some cases, ROS signals can elicit a hypersensitive reaction during symbiosis that is probably involved in the autoregulation of nodulation [12,13]. Therefore, rhizobia must tolerate ROS stress before and after symbiosis.

*Azorhizobium caulinodans* can establish a symbiotic interaction with the tropical legume *Sesbania rostrata*, and it forms nodules on its roots and stems [14]. In general, the stem nodules are completely exposed to the air, and the cortex cells contain rich chloroplasts [15]. The chloroplast is a major source of ROS production in plants [16]. Thus, rhizobia in stem nodules are more susceptible to ROS. Consequently, *A. caulinodans* has evolved several types of ROS scavenging systems for sensing, regulating, and protecting against ROS toxicity from the host. In our previous work, we have comprehensively studied several antioxidant enzymes in *A. caulinodans*, such as catalase (KatG) [17], alkyl hydroperoxide reductase (AhpC) [18], and bacterioferritin comigratory protein (BCP) [19], which are all involved in detoxifying exogenous H_2_O_2_, and affect the nodulation behaviour and nitrogen fixation in *S. rostrata* [17,18,19]. However, plants produce additional substances as part of the active defence response, such as organic hydroperoxides (OHPs) [20,21]. It has been noted that OHPs are more lethal and they react with cell membranes to generate toxic ROS [22]. However, the pathways protecting against OHPs in *A. caulinodans* remain unclear.

Bacteria protect themselves against OHPs toxicity via two mechanisms. The first involves the well-characterised enzyme AhpC, which is a member of the peroxiredoxin family and it degrades both organic and inorganic peroxides [23,24,25]. However, our previous study revealed that, in *A. caulinodans*, AhpC was involved in the detoxification of H_2_O_2_, but not OHPs [18]. The other enzyme that is involved in the detoxification of OHPs is the organic hydroperoxide resistance protein (Ohr), first discovered in *Xanthomonas campestris* [26], which was later revealed to be widely distributed in bacteria [27,28,29]. Ohr is a type of thiol-dependent peroxidase that catalyses the reduction of organic peroxides into corresponding alcohols [26,30]. Studies in several types of bacteria revealed that Ohr more effectively reduces OHPs than H_2_O_2_ [27,28,31]. Moreover, an *ohr* mutant strain was found to be highly sensitive to OHPs, but not to H_2_O_2_ [22,29,32]. Ohr has less similarity to OsmC (osmotically inducible protein), another thiol-dependent peroxide reductase that responds to osmotic stress [33,34]. Despite the fact that both Ohr and OsmC are structurally and functionally homologous proteins, they display different patterns of regulation [35,36,37]. 

OhrR, a member of the MarR family, regulates the expression of *ohr* [27,38,39]. Members of this family act as dimeric proteins that sense and modulate resistance against several cellular toxins, including antibiotics, detergents, and ROS [40,41,42]. Biochemical and structural data showed that reduced OhrR functions as a dimeric repressor that binds the inverted repeated sequences in the *ohr* promoter, thereby inhibiting transcription [27,43]. OHPs-induced oxidative stress results in the oxidation of OhrR, which subsequently undergoes a conformational modification that decreases its affinity to DNA, leading to the expression of *ohr* [43,44]. OhrR harbours a conserved cysteine residue in its N-terminal region that senses OHPs via several mechanisms of redox regulation [44]. OhrR proteins are classified into two subfamilies that are based on the number of cysteine residues: the 1-Cys subfamily, which contains a single conserved cysteine that is best characterized in *Bacillus subtilis*, and the 2-Cys subfamily, an example of which is the OhrR in *X. campestris*, which contains two redox-active cysteine residues [45,46].

Ohr and OhrR have been studied in several types of bacteria; however, little is known regarding the role of these proteins in rhizobia. Although our previous study investigated the H_2_O_2_ scavenging systems in *A. caulinodans*, the OHPs resistance network remains poorly understood [17,18,19]. The present study scrutinized the functional role of the *ohr* and *ohrR* genes in *A. caulinodans* ORS571. We investigated the biochemical properties of *ohr*/*ohrR* system and their effect on symbiotic behaviours of *A. caulinodans* ORS571. Furthermore, the regulation pattern of Ohr and OhrR was also studied in vitro and in vivo.

## 2. Materials and Methods 

### 2.1. Bacterial Strains, Plasmids and Cultural Conditions

Appendix A lists bacterial strains and plasmids in this work. *Escherichia coli* strains were grown in Luria–Bertani (LB) broth at 37 °C [47]. *Azorhizobium caulinodans* ORS571 and its derivative strains were grown in tryptone–yeast extract (TY) medium at 28 °C [48]. All of the solid media contained 1.2% agar. The indicated antibiotics were added for selection at final concentrations, as following: ampicillin (Amp, 100 µg·mL^−1^), gentamicin (Gm, 20 µg·mL^−1^), kanamycin (Km, 100 µg·mL^−1^), and spectinomycin (Sm, 100 µg·mL^−1^). Bacterial growth was determined by measuring OD_600_ using spectrophotometer (Eppendorf, Hamburg, Germany).

### 2.2. Bioinformatics Analyses

Genome sequences were downloaded from the NCBI gene database (www.ncbi.nlm.nih.gov/gene). The sequence of SmOhr and SmOhrR in *Sinorhizobium meliloti* was used to identify similar sequences of Ohr and OhrR in *A. caulinodans* by the BLASTP program in National Center for Biotechnology Information (NCBI: https://www.ncbi.nlm.nih.gov) [22,49]. Multiple sequence alignments that were based on protein sequences were constructed using the MegAlign and GeneDoc alignment software.

### 2.3. Construction of In-Frame Deletion and Complementation

*A. caulinodans* ORS571 was used as the parental strain for generating the in-frame deletion of *ohr* (AZC_2977) and *ohrR* (AZC_3555) following the previously described method [17]. Brifiely, the flanking fragments of gene *ohr* and *ohrR* were cloned into a suicide vector pEX18Gm containing the *sacB* gene [50]. Appendix A lists the primers used to generate upstream and downstream regions. Appendix A shows schemes of plasmids construction and the recombination process. The constructed plasmids pEX18Gm-ohr/ohrR were introduced into *E. coli* SM10 λpir to serve as the donor in the biparental conjugation, with strain *A. caulinodans* ORS571 as the recipient. For biparental conjugation, the donor and recipient cells were combined at 1:3 based on optical density values. The mixed cell solution was loaded on a filter membrane placed on TY agar. After incubation at 28 °C for 12 h., the conjugation colonies were selected on TY plates supplemented with ampicillin and gentamicin. Double-crossover events were selected on TY-sucrose (10% sucrose) plates after the first “cross-in” homologous recombination The positive colonies were selected and streaked on TY plates containing Amp or Amp/Gm at 28 °C for three days. The colonies that only grew on TY plates containing Amp were further confirmed by sequencing. For complementation analysis, the coding regions of *ohr* and *ohrR* were amplified and cloned into the plasmid pYC12 by PCR while using the primers that are listed in Appendix A [51]. The resulting recombination plasmids were conjugated into its corresponding deletion mutant strain by electroporation.

### 2.4. Disc Diffusion Assay

Disc diffusion assay was performed to determine the resistance of *A. caulinodans* strains to organic peroxides according the previously described method [52]. Cumene hydroperoxide (CuOOH), a typical organic peroxide, was used in this assay [53]. Approximately 10^7^ bacterial cells of *A. caulinodans* strains were mixed with the semi-solid TY agar (0.6%) and then poured onto TY agar plates. 4 μL of 4.88 M CuOOH was loaded on 6 mm sterilised paper disks (Waterman) that were placed on top agar. The plates were incubated for 48 h at 28 °C and the inhibition zone was measured and adjusted by subtracting the diameter of the paper discs. The experiment was repeated at least three times.

### 2.5. Peroxide Killing Assay

Various concentrations of CuOOH were used to test the sensitivity of *A. caulinodans* strains to organic peroxides by killing assays [17]. The *A. caulinodans* strains were grown to early-log phase (OD_600_ ≈ 0.2) in TY liquid medium, CuOOH was added to cultures at various final concentration. After incubation for 4 h, a series of 10-time dilutions of treated cultures were placed onto TY agar plates. The CFU (colony forming units) was determined after incubation at 28 °C for three days. This experiment was repeated three times.

### 2.6. Quantitative Real-Time PCR Analysis

qRT-PCR monitored the transcriptional level of gene *ohr* regulated by *ohrR*. The total RNA were exacted from *A. caulinodans* strains by TRIzol method [54]. Reverse transcription PCR was performed using a cDNA Synthesis Kit (Vazyme Biotech, Nanjing, China), according to the manufacturer’s instructions. Appendix A lists specific primers for qRT-PCR amplification. The qPCR program was as follows: 30 s at 95 °C, followed by 40 cycles of 10 s at 95 °C and 30 s at 60 °C, according to the manufacturer’s protocol for SYBR green detection (Vazyme Biotech, Nanjing, China). The quantification of gene expression and melting curve analysis were completed using a 7500 Plus Real-Time PCR System (Applied Biosystems, Foster City, CA, USA). An endogenous control (16S rRNA) was used for signal normalization [55]. All of the experiments were conducted in at least three independent replicates, and the relative expression levels of target genes were calculated following the Comparative CT method [56].

### 2.7. Translational Analysis of ohr

A *lacZ*-based reporter system was used to characterise the gene expression of *ohr* in *A. caulinodans* WT and Δ*ohrR* strains. DNA fragment covers the promoter region of *ohr* gene (from -453 to +47, the nucleotide positions are numbered from the first position of the initiation codon of the ohr gene) was cloned into a translational fusion plasmid vetor pRA302 and then transformed into WT and Δ*ohrR* strains [57]. Appendix A lists the primers used for construction. For the induction of *lacZ* expression, CuOOH was added at the early-log stage. After incubation for 2 h, the cells were collected for measuring the β-galactosidase activities following the previously described method [58]. This experiment was performed three times.

### 2.8. Protein Expression and Purification

The open reading frame of gene *ohrR* was cloned into the expression vector pET-28a (Novagen, Madison, WI, USA) and then transferred into *E. coli* BL21(DE3) (Appendix A). The recombinant strain was cultured in 1L LB medium that was supplemented with 100 µg·mL^−1^ kanamycin at 37 °C. Protein expression was induced at OD_600_ ≈ 0.8 by the addition of isopropyl β-d-1-thiogalacto-pyranoside (IPTG) with a final concentration of 0.5 mM. The induced cultures were then grown at 16 °C for 12 h. The recombinant protein carried His-tag at the N-terminal and was purified by Ni-NTA agarose (GE Healthcare, Piscataway, NJ, USA). Protein purification was analyzed by non-reducing SDS-PAGE. The purified protein was desalted by HiTrap Desalting Column (GE Healthcare, Piscataway, NJ, USA). CuOOH was added to the binding buffer at a final concentration of 0.1 mM, and then incubated for 10 min. to oxidize OhrR protein. Restored OhrR protein was performed by incubating OhrR with DTT (final concentration: 0.5 mM) for 10 min.

### 2.9. Electrophoretic Mobility Shift Assay

The binding of OhrR protein or its mutated derivatives to target promoters was performed, as described previously [59,60]. The promoter region of *ohr* was amplified and purified as DNA probe (181 bp). EMSA was performed by adding increasing amounts of purified OhrR protein to the DNA fragment (final concentration: 20 nM) in a total volume of 20 μL. The binding buffer contained 50 mM Tris-HCl (pH 8.3), 0.25 M KCl, 2.5 mM DTT, 5 mM MgCl_2_, 0.05 µg·mL^−1^ poly(dI-dC), 2.5 mM EDTA, 1% glycerol. The reaction mixtures were incubated for 20 min. at room temperature and then loaded onto a 6% native polyacrylamide gel in 0.5× Tris-borate-EDTA buffer at 150 V for 70 min. The gel was subsequently stained with GelRed (Sangon Biotech, Shanghai, China) for 20 min. and then imaged while using the gel imaging system. 

### 2.10. Site-Directed Mutagenesis

OhrR cysteine site-directed mutagenesis was performed by overlap PCR according to the method used before [61]. The plasmid pET28a-OhrR was used as template to generate the mutants by substituting cysteine for serine, respectively. Appendix A lists the complementary mutagenic primers. The *ohrR* fragments containing mutations were cloned into the expression vector pET-28a, resulting in pET-28a-OhrR^C11S^, pET-28a-OhrR^C121S^, and pET-28a-OhrR^C11SC121S^, all of which were transferred to *E. coli* BL21(DE3) for in vitro assay. To produce mutated OhrR in vivo, all mutated *ohrR* genes were, respectively, cloned into pYC12, resulting in pYC12-OhrR^C11S^, pYC12-OhrR^C121S^, and pYC12-OhrR^C11SC121S^. The recombinant plasmids were introduced into *ohrR* mutant strain by conjunction. All of the mutagenesis were confirmed by sequencing.

### 2.11. Nodulation Assay and Nitrogenase Activity

The *Sesbania rostrata* seeds were surface sterilized and grown in sterilized vermiculites, as described previously [62]. Overnight cultures of *A. caulinodans* WT and Δ*ohr* were diluted to OD_600_ values of 1.0 (approximately 10^8^ cells) for inoculation. Bacterial suspension was spread onto the stem of *Sesbania rostrata* for stem nodulation (six biological replicates (individual plants) for each strain). Stem nodules were harvested to count numbers after 35 days post-inoculation. For root nodulation on *S. rostrata*, seeds were immersed in approximately 10^8^
*A. caulinodans* cells. After 20 min, the seedlings were transferred to pots containing sterilized vermiculites and grown in a plant growth chamber at 28 °C [55]. N_2_-fixing activity was measured by the acetylene reduction assay (ARA test), as reported previously [63]. Fifteen nodules per plant were placed into a 20 mL vial sealed by a butyl rubber septum. Each vial was injected with 2 mL acetylene by a syringe and then incubated at 28 °C for 2 h. The ethylene production was detected by using an HP 6890 Series Gas Chromatograph System (Agilent, Palo Alto, CA, USA). Gas chromatography was conducted to measure the peak height of ethylene and acetylene with 100 μL of gas. The approximate nitrogenase activity was expressed as μmol of C_2_H_4_ of acetylene production per gram of nodule dry weight.

## 3. Results

### 3.1. Identification of A. caulinodans Ohr and OhrR

Bioinformatics analyses of the *A. caulinodans* ORS571 genome were performed in order to study the organic peroxide resistance system in *A. caulinodans*. We identified the putative gene encoding Ohr (AZC_2977) as a member of the OsmC/Ohr family [14]. AZC_2977 located in the region between 3,402,526 and 3,402,951 base pairs of the *A. caulinodans* genome, theoretically encoding a 14.3-kDa protein. AZC_2977 shares 62, 52, 64, and 67% amino acid sequence identity with the Ohr proteins in *X. campestris* (GenBank Accession Number: AAC38562), *Pseudomonas aeruginosa* (NP_251540), *Agrobacterium tumefaciens* (NP_353869), and *Sinorhizobium meliloti* (NP_3835068), respectively (Figure 1A,B) [22,26,64,65]. We also found that AZC_3555, located between base pairs 4,083,119 and 4,083,553 of the *A. caulinodans* genome, encodes an amino acid sequence that has a high similarity to OhrR proteins in *B. subtilis* (47%) (O34777), *X. campestris* (53%) (AAK62673), *P. aeruginosa* (46%) (AE004711_9), and *S. meliloti* (56%) (NP_385067) (Figure 1A,C) [22,27,38,65]. The predicted molecular weight of the AZC_3555 protein in *A. caulinodans* was 16.1 kDa. Additionally, AZC_3555 contains two conserved cysteine residues that were found in all of the investigated 2-Cys OhrR proteins, at position 11 (Cys_11_) and position 121 (Cys_121_), which are thought to activate the protein via oxidation. AZC_2977 and AZC_3555 were subsequently termed Ohr and OhrR in *A. caulinodans*, respectively. 

### 3.2. Ohr Contributes to Defence against Organic Peroxides in A. caulinodans

In several types of bacteria, genes that are divergently transcribed from the *ohr* gene often encode the transcriptional repressor OhrR [22,38,66]; however, this is not the case in *A. caulinodans*. We generated in-frame deletion mutants of *ohr* and *ohrR*, and monitored the growth behaviour of the wild-type (WT) and mutant strains to determine the role of Ohr and OhrR in the OHPs resistance network in *A. caulinodans*. The growth curves of the WT and mutant strains showed that there was no significant difference between these strains in the absence of CuOOH (Appendix A), indicating that Δ*ohr* and Δ*ohrR* do not affect growth under normal conditions. A disk diffusion plate assay was used to determine the resistance of Δ*ohr* and Δ*ohrR* mutants to CuOOH (4.88 M). The diameter of the inhibition zone of Δ*ohr* (54.0 ± 1.5 mm) was shown to be significantly increased when compared with that of the WT strain (30.3 ± 0.5 mm), which indicated that Ohr played a critical role in the defence against OHPs in *A. caulinodans* (Figure 2A,B). Complementation of Δ*ohr* partially restored the phenotype to that of the WT strain (36.7 ± 1.8 mm). By contrast, the inhibition zone of Δ*ohrR* was similar to that of the WT strain (30.1 ± 1.2 mm), suggesting that the deletion of *ohrR* did not affect the resistance against OHPs (Figure 2A,B) because it normally acts as an inhibitor [44]. We subsequently performed a killing assay to test the sensitivity of WT and mutant strains to different levels of CuOOH. When exposed to 0.3 mM CuOOH, Δ*ohr* exhibited hypersensitivity to CuOOH as compared with the other strains, as shown in Figure 2C. In the presence of 0.5 mM CuOOH, the viability of all the tested strains was suppressed. The survival rate of Δ*ohrR* was not distinguishable from that of the WT strain. Collectively, these data revealed the importance of Ohr in OHPs resistance in *A. caulinodans*.

### 3.3. The Expression of ohr Is Regulated by OhrR

OhrR was reported to repress *ohr* in several types of bacteria [64,65]. qPCR and an *ohr-lacZ* translational fusion reporter plasmid were used to measure *ohr* expression in WT and ∆*ohrR* strains in order to evaluate this effect in *A. caulinodans*. We found that the transcription of *ohr* was significantly increased (~6-fold) in the ∆*ohrR* strain as compared with the WT strain in the absence of CuOOH (Figure 3A), which was consistent with OhrR functioning as a repressor [39]. In the presence of CuOOH, a substantial increase in *ohr* expression in the WT strain was observed compared with untreated samples (Figure 3A). No differences were observed in the ∆*ohrR* strain under the two conditions. These data demonstrated that CuOOH induced the expression of the *ohr* gene by removing the inhibitory effect of OhrR. The translational reporter system test corroborated this result. We found that the expression of *ohr* was upregulated in the absence of *ohrR* under normal conditions (Figure 3B), consistent with OhrR functioning as a repressor. In the presence of CuOOH, the expression of *ohr* was similar in the *ohrR* mutant and WT strains (Figure 3B). These results are in accordance with the repression of *ohr* by OhrR in the absence of CuOOH.

### 3.4. Reduced Form of OhrR Binds the Promoter Region of ohr and This Binding Is Inhibited by Peroxides

Electrophoretic mobility shift assay was used to investigate the binding interaction between OhrR and *ohr* promoter. Appendix A shows the sequence of *ohr* promoter, an inverted repeat motif was found in the promoter region. OhrR was expressed in *E. coli* as a 6×His-tagged fusion protein, with the tag at the N-terminus. Recombinant OhrR protein (molecular weight, ~17 kDa) was purified while using a Ni-NTA resin column (Figure 4A). The incubation of the OhrR protein with 0.1 mM CuOOH led to the formation of a covalent dimer (~35 kDa). The formation of the dimer was reversed by 0.5 mM DTT treatment (Figure 4A). Thus, the oxidation of OhrR induced a reversible bond between the two protein subunits. EMSA was performed with purified OhrR protein and DNA fragment covering the promoter region of *ohr*. A protein-DNA complex was observed after the interaction of the OhrR protein with the target DNA, which indicated that the OhrR protein binds to the *ohr* promoter in a dose-dependent manner (Figure 4B). As controls, heat-inactivated OhrR did not bind to the *ohr* promoter (lane 7), and active OhrR did not bind to a nonspecific promoter (16S rDNA) (lane 8). The binding of OhrR to the *ohr* promoter region was suppressed in the presence of CuOOH, and recovered upon the addition of DTT (Figure 4B), which suggested that only the reduced form of OhrR was able to bind DNA. Collectively, these data indicated that the OhrR protein in *A. caulinodans* controls *ohr* by directly binding to the promoter region, while the presence of OHPs restricts this binding. 

### 3.5. Sensing of Organic Hydroperoxides by OhrR Requires the Conserved Cys11 Residue

Sequence alignment showed that the OhrR from *A. caulinodans* contains two conserved cysteines, at position 11 (Cys_11_) and position 121 (Cys_121_) (Figure 1A), and it is a typical OhrR protein belonging to the 2-Cys subfamily [43]. Site-directed mutagenesis was used to generate OhrR^C11S^, OhrR^C121S^, and OhrR^C11SC121S^ proteins, in which cysteine residues were replaced by serine, to determine whether these cysteine residues are involved in redox sensing in vitro. All of the recombinant proteins were expressed, purified, and subjected to EMSA with the *ohr* promoter. Under reducing condition, the three mutant OhrR were able to bind the *ohr* promoter (Figure 5A). The addition of CuOOH abolished the binding activity of OhrR^C121S^. However, the DNA-binding activity of OhrR^C11S^ and OhrR^C11SC121S^ was not affected by CuOOH treatment (Figure 5A). The non-reducing SDS-PAGE result of OhrR variants showed that the OhrR^C11S^ and OhrR^C11SC121S^ proteins remained predominantly as monomers under the CuOOH exist condition, Conversely, a dimerization upon CuOOH oxidation was observed for the OhrR^C121S^ protein, which suggested that Cys_121_ has less role in dimer formation (Appendix A). An in vivo assay showed that *in trans* expression of these mutant proteins in the Δ*ohrR* strain reduced the resistance to CuOOH, whereas the effect of the Cys_11_ substitution was stronger than Cys_121_ substitution (Figure 5B), which supported the hypothesis that the conserved Cys_11_ is the key residue responsible for the OHPs sensing ability of OhrR. 

### 3.6. Ohr Is Required for Optimal Stem Nodulation and Nitrogenase Activity

We measured nitrogenase activity in stem nodules produced by WT and Δ*ohr* strains to investigate the role of Ohr in the symbiotic relationship. Approximately 10^8^ cells of WT and Δ*ohr* cells were sprayed onto the stems of *S. rostrata*. After 35 days, we found that the WT and Δ*ohr* strains both induced well-developed nodules on the stems of *S. rostrata*; however, Δ*ohr* formed significantly fewer nodules than the WT (Figure 6A,B). Furthermore, there was no significant difference between WT and Δ*ohr* in root nodules formation (Appendix A). We collected the stem nodules to test whether Ohr affected nitrogen fixation efficiency. The stem nodules inoculated with the Δ*ohr* strain exhibited substantially reduced nitrogenase activity as compared with those inoculated with the WT strain, as shown in Figure 6C. These data suggested that Ohr is important for nodulation and nitrogen fixation. 

## 4. Discussion

*A. caulinodans* and other rhizobia must be able to withstand oxidative stress in the host environment to establish successful symbiotic relationships with the host plants. At the first encounter with *S. rostrata*, *A. caulinodans* is recognised as an invader and triggers an oxidative burst [11,18,67]. OHPs are produced by host plants as a part of the defence response against bacterial infection, and subsequently react with free fatty acids and cell membranes, leading to the production of other organic radicals, which increases their toxicity [26].

The present study focused on lipoyl-dependent peroxidase expression and regulation through the genes *ohr* and *ohrR*, and investigated their contribution to ROS resistance in *A. caulinodans*. We identified and characterised the essential role of *ohr* and *ohrR* in *A. caulinodans* defence mechanisms against OHPs. The Δ*ohr* strain appeared to be hypersensitive to CuOOH, and this sensitivity was nearly restored to WT levels following complementation with the *ohr* gene (Figure 2A,B). These results suggest that Ohr plays a vital role in the detoxification of OHPs in *A. caulinodans*, which is consistent with the role of Ohr in other organisms [22,26,27,28]. However, the deletion of the *ohrR* gene did not impact the sensitivity to CuOOH (Figure 2A,B), and the *ohrR* mutant exhibited a survival rate comparable to the WT strain in the presence of CuOOH (Figure 2C). This suggested OhrR might repress *ohr* expression, and that CuOOH functions as an inducer. OHPs are principally degraded by the Ohr system in *A. caulinodans,* as AhpC and BCP are involved in the scavenging of H_2_O_2_ but not OHPs [18,19]. 

The regulation pattern of *ohr* was examined at the transcriptional and translational levels. The results showed that the deletion of *ohrR* increased the expression of *ohr* in the presence or absence of CuOOH (Figure 3A,B). These data indicated that the expression of *ohr* is inhibited by OhrR. The EMSA results showed that reduced OhrR (monomer) could bind to the *ohr* promoter and repress its transcription, potentially by competitively binding to the binding site to block the access of RNA polymerase [68]. We observed that OhrR oxidation led to the formation of a dimer that detaches from the target DNA, and this phenomenon was reversed by DTT in vitro (Figure 4A). This result explained the similarity of the phenotype of the strain with constitutive *ohrR* expression to that of the WT strain in the presence of CuOOH (Figure 2A,B). In most bacteria, *ohr* and *ohrR* located in the same operon, *ohr* promoter is located within the *ohrR*-*ohr* intergenic region. In the presence of OHPs, the monomeric OhrR represses the expression of both *ohr* and *ohrR* at the transcriptional level through binding to the *ohr*-*ohrR* intergenic region [22]. However, *ohr* and *ohrR* located separately on the genome of *A. caulinodans*, suggested a differential regulation mechanism existed. Sequencing alignment results showed that OhrR in *A. caulinodans* belongs to the 2-cysteine subfamily of OhrR proteins (Figure 1C) [46]. The conserved Cys_11_ allows for OhrR to sense OHPs, as evidenced by EMSA (Figure 5A,B), while Cys_121_ appears to have a role in preventing overoxidation. It is speculated that, in the presence of OHPs, Cys_11_ initially reacts with OHPs, producing a sulfenic acid intermediate that is subsequently attacked by Cys_121_, leading to the formation of an intramolecular disulphide bond between Cys_11_ and Cys_121_ [69]. These results indicate that *A. caulinodans* displays complex patterns of transcriptional changes that accompany the adaptation to host OHPs stress. Recently, OhrR was also shown to control the expression of virulence genes in *Vibrio cholerae* and *Burkholderia thailandensis* [70,71], regulate the production of bactericides in *Streptomyces avermitilis* [72], and mediate the response to hydroperoxides in *Shewanella oneidensis* [32]. These studies suggest that OhrR might participate in multiple mechanisms of resistance regulation and further investigation is required to determine the role of OhrR in *A. caulinodans*.

In *A. caulinodans*, the deletion of *ohr* significantly affected stem nodule formation in *S. rostrata* (Figure 6A,B). Moreover, the stem nodules formed by the *ohr* mutant showed lower nitrogen fixation activity than the WT stem nodules (Figure 6C). In *S. meliloti*, the inactivation of *ohr* did not affect symbiosis and nitrogen fixation [22]. However, *ohr* is demonstrated to be involved in the nodulation and nitrogen-fixing activity in *A. caulinodans*. Plants generate OHPs as a defence mechanism during the early phase of symbiosis. It is possible that Ohr reduces OHPs, allowing for *A. caulinodans* to invade *S. rostrata* via crack entry [73]. Stem nodules contain photosynthetic chloroplasts that produce oxygen, whereas nitrogenase is extremely sensitive to oxygen and might be inactivated even at low oxygen concentrations [74]. The deletion of *ohr* did not affect root nodule formation, which suggests that the oxygen microenvironment differs between stem and root nodules, affecting the physiological function of *A. caulinodans* [15]. In the process of nodulation and nitrogen fixation, *A. caulinodans* encounters several types of ROS. Therefore, multiple ROS scavenging systems are required for *A. caulinodans* to establish optimal symbiosis with *S. rostrata*. Our work contributes to the elucidation of the molecular bases of ROS signals network during symbiotic interactions between *A. caulinodans* and *S. rostrata*. However, further in-depth research is required to verify the cross-talk between different ROS resistant systems that are involved in the interactions between rhizobia and legumes.

## Figures and Tables

**Figure 1 genes-11-00335-f001:**
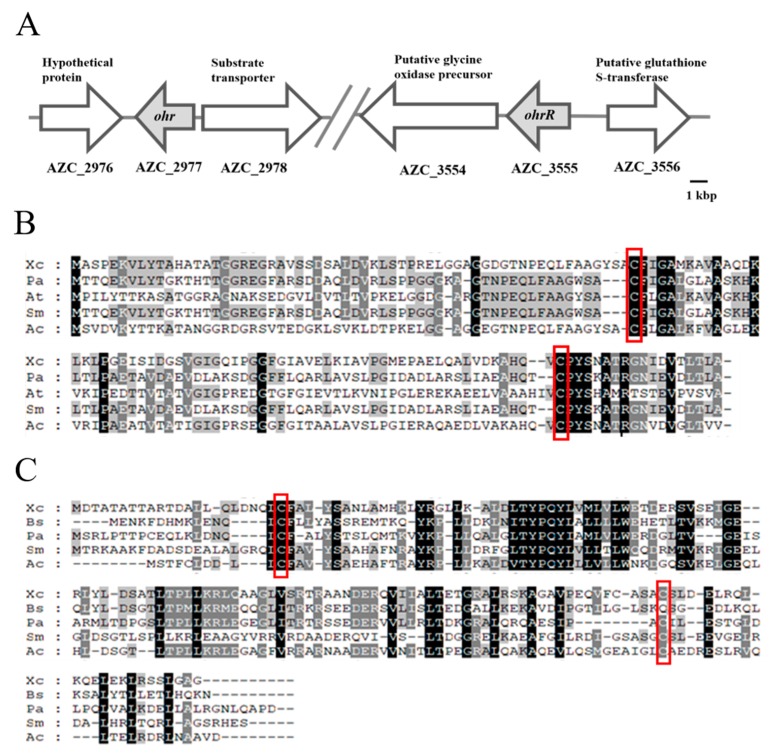
Identification of organic peroxide relevant genes *ohr* and *ohrR* in *Azorhizobium caulinodans* ORS571. (**A**) Genomic organization of the *ohr* and *ohrR* region in *A. caulinodans* ORS571. (**B**) Alignment of the amino acid sequence of Ohr from *Xc* (*Xanthomonas campestris*), *Pa* (*Pseudomonas aeruginosa*), *At* (*Agrobacterium tumefaciens*), *Sm* (*Sinorhizobium meliloti*), and *Ac* (*Azorhizobium caulinodans* ORS571). (**C**) Alignment of the amino acid sequence of OhrR from *Xc*, *Bs* (*Bacillus subtilis*), *Pa*, *Sm,* and *Ac*. Red rectangular boxes indicate conserved cysteine residues.

**Figure 2 genes-11-00335-f002:**
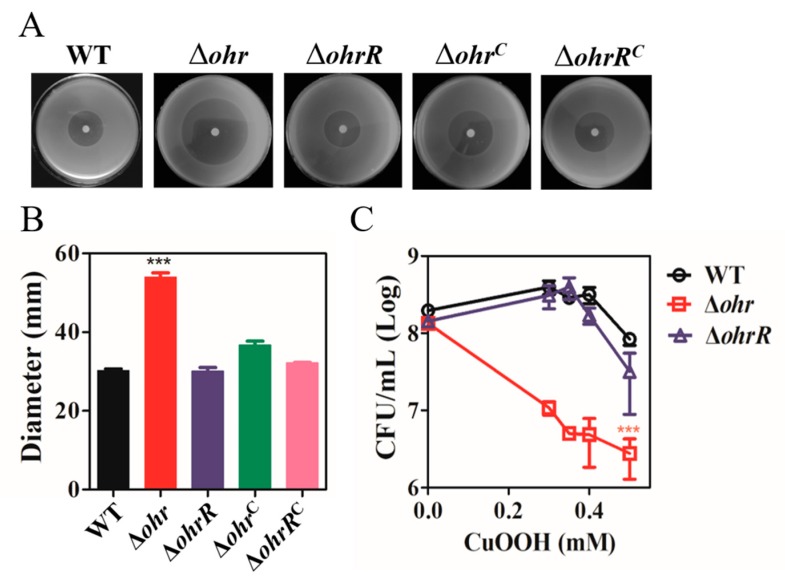
Characteristics of the *A. caulinodans* mutants in response to CuOOH. (**A**,**B**) Disk diffusion assay. Approximately 10^7^ bacterial cells of *A. caulinodans* were mixed and spread on TY agar plates. Paper discs of 6 mm in diameter loading with 4 μL CuOOH of 4.88 M were placed on bacterial lawns. Plates were incubated at 28 °C for two days until measurement of inhibition zones. (**C**) Killing assay. CuOOH was added to early-log cultures to the final concentrations as indicated. After incubated for 4 h, samples were properly diluted and plated on TY plates. Colony counting was done after 3 days. Data are mean and SD of three independent experiments. ***: Student *t*-test *P* < 0.005, ns: no significant difference.

**Figure 3 genes-11-00335-f003:**
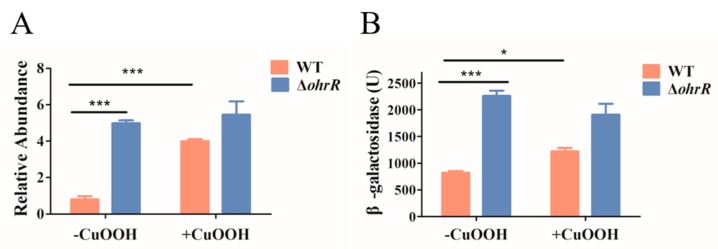
Induction of the expression of *ohr* and *ohrR* by organic peroxides. (**A**) Transcription of the *ohr* gene. qRT-PCR analysis of RNA extracted from mid-log growing cells that were untreated and treated with 0.1 mM CuOOH. The expression of *ohr* was normalized with 16S rRNA. (**B**) Translational fusion assay. *ohr-lacZ* gene expression was analyzed by measuring β-galactosidase levels. Overnight cultures of WT and Δ*ohrR* strains containing *ohr-lacZ* plasmids were incubated to early-log phase. When indicated, 0.1 mM CuOOH was added and all of the cultures were incubated for 2 h. The *ohr* gene expression was analyzed by measuring β-galactosidase activities. Data are mean and SD of three independent experiments. *: Student *t*-test *P* < 0.05; ***: *P* < 0.005.

**Figure 4 genes-11-00335-f004:**
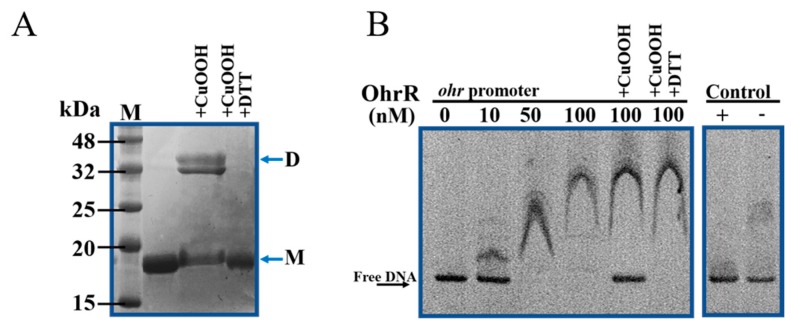
OhrR binds with high affinity to the promoter region of *ohr*. (**A**) Purified OhrR treated with CuOOH or DTT was analyzed by non-reducing SDS-PAGE. The monomeric (M) and dimeric (D) form of OhrR are indicated by arrows. (**B**) EMSA was performed to detect the interaction of OhrR with the *ohr* promoter. DNA fragment containing the promoter region of *ohr* was incubated with increasing concentrations of purified OhrR protein, as indicated, and the mixtures were separated in a polyacrylamide non-denaturing gel. CuOOH was added to the reaction (final concentration: 0.1 mM), and incubated for 10 min., when indicated, added with DTT (0.5 mM) and incubated for another 10 min.

**Figure 5 genes-11-00335-f005:**
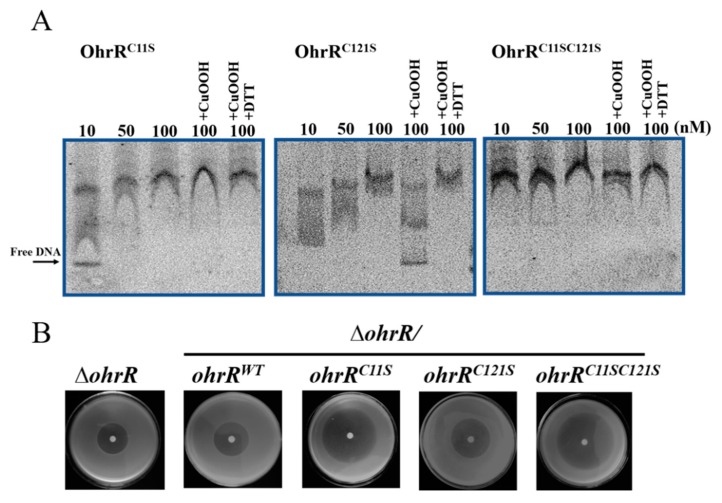
Effect of OhrR cysteine mutations on peroxide sensing in vitro and in vivo. (**A**) EMSA. DNA fragment covering the promoter region of *ohr* was incubated with the given concentration of purified OhrR^C11S^, OhrR^C121S^, OhrR^C11SC121S^ proteins. 0.1 mM CuOOH or 0.1 mM CuOOH plus 0.5 mM DTT were added to the mixtures. (**B**) Disc diffusion assay. Paper discs of 6 mm in diameter loading with 4 μL 4.88 M CuOOH were placed on bacterial lawns. The inhibition zones were observed around the disks after incubated at 28 °C for 48 h.

**Figure 6 genes-11-00335-f006:**
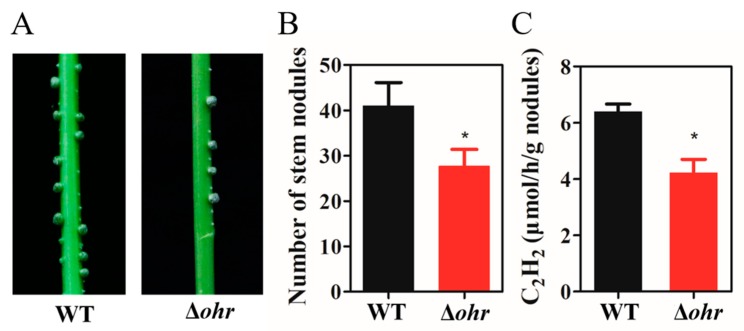
Symbiotic performance of *A. caulinodans* WT and Δ*ohr* on *S. rostrata*. (**A**) Pictures of stem nodules formed by WT and Δ*ohr* after 35 days post-inoculation. (**B**) The number of stem nodules on *S. rostrata*. (**C**) Nitrogen fixation activities of stem nodules. The values are means and standard deviations from six replicate plants. *: Student *t*-test *P* < 0.05.

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
