# Peer review of "Ohr and OhrR Are Critical for Organic Peroxide Resistance and Symbiosis in Azorhizobium caulinodans ORS571"

_genes, 2020, doi:10.3390/genes11030335_

Round 1

Reviewer 1 Report

The paper by Guo et al. shows the role of ohr and ohrR genes in the resistance to organic peroxides  and in symbiosis of Azorhizobium caulinodans. The regulation of ohr was also evaluated at transcriptional and translational levels and OhrR binding to the ohr promoter has been demonstrated.

The work is interesting and brings new results on the regulation and function of the hydroperoxide resistance protein Ohr of A. caulinodans, an endosymbiont of Sesbania rostrata.

Comments, suggestions and information that could be included

Cumene hydroperoxide (CuOOH). Perhaps a minimum reference to the use and characteristics of this compound could be included

Line 107. For the complementation, it is explained that the coding regions have been used but the promoter regions included should be detailed.

Line 135, could you indicate the nucleotide number of the cloned fragment, and and if it covers the entire intergenic region that precedes the ohr gene?. To understand it better, in Fig 1A, either genes adjacent to AZC_2977 and AZC_3555 could be added. Perhaps it would also be interesting to compare the ohr / ohrR gene organization of the bacteria in Figure 1.

ohr Promoter. Does A. caulinodans ohr promoter  contain inverted repeated sequences? Can be showed in a Fig?. and the ohrR promoter? This information could be included in the manuscript

Line 151. What is the sequence and size of the promoter used?

Line 237. In Fig 3. Color or texture of the columns should be changed in order to better appreciatte the differences. Units have to be indicated.

Line 249. OhrR and the ohr promoter

It can be mentioned that the OhrR contained a histidine tag (OhrR-His)

Line 256, controls are not shown?

Line 263. Fig. 4A, lanes 1 and 2 are the same?

Do OhrRC11S, OhrRC121S and OhrRC11SC121S form dimers? This information could be at Fig4A and would allow to discuss if C11 is important for dimerization.

Line 290, title should be changed, Ohr is required for optimal stem nodulation….????

Line 301. How many nodulation assays have been done to obtain the results of fig 6B and C.  And in Fig S1 and Fig. S3? 

Lines 18, 49, 57, 294, 303, 343, 347, 353 Sesbania rostrata

Author Response

Reviewer #1:

The paper by Si et al. shows the role of ohr and ohrR genes in the resistance to organic peroxides and in symbiosis of Azorhizobium caulinodans. The regulation of ohr was also evaluated at transcriptional and translational levels and OhrR binding to the ohr promoter has been demonstrated.

The work is interesting and brings new results on the regulation and function of the hydroperoxide resistance protein Ohr of A. caulinodans, an endosymbiont of Sesbania rostrata.

Comments, suggestions and information that could be included

  1. Cumene hydroperoxide (CuOOH). Perhaps a minimum reference to the use and characteristics of this compound could be included.

Author reply: Added. Reference 53.

  1. Line 107. For the complementation, it is explained that the coding regions have been used but the promoter regions included should be detailed.

Author reply: Added and Fig S3.

  1. Line 135, could you indicate the nucleotide number of the cloned fragment, and if it covers the entire intergenic region that precedes the ohr gene? To understand it better, in Fig 1A, either genes adjacent to AZC_2977 and AZC_3555 could be added. Perhaps it would also be interesting to compare the ohr / ohrR gene organization of the bacteria in Figure 1.

Author reply: Added. Fig 1A was modified.

  1. ohr Does A. caulinodans ohr promoter contain inverted repeated sequences? Can be showed in a Fig? and the ohrR promoter? This information could be included in the manuscript.

Author reply: Thank you for your suggestion. And we add the information in Fig S3.

As for the promoter of OhrR, we did not find any inverted repeated sequence.

  1. Line 151. What is the sequence and size of the promoter used?

Author reply: The length of DNA fragment used in EMSA is 181 bp, covering the promoter region of ohr. The promoter sequence is shown in supporting information (Figure S3).

  1. Line 237. In Fig 3. Color or texture of the columns should be changed in order to better appreciate the differences. Units have to be indicated.

Author reply: Changed in figure 3

  1. Line 249. OhrR and the ohr promoter.

Author reply: Corrected.

  1. It can be mentioned that the OhrR contained a histidine tag (OhrR-His).

Author reply: Added.

  1. Line 256, controls are not shown?

Author reply: Corrected. Controls are shown in Figure 4B, lane 7 and lane 8.

  1. Line 263. Fig. 4A, lanes 1 and 2 are the same?

Author reply: Corrected.

  1. Do OhrRC11S, OhrRC121S and OhrRC11SC121S form dimers? This information could be at Fig4A and would allow to discuss if C11 is important for dimerization.

Author reply: Corrected and discussed in part 3.5

  1. Line 290, title should be changed, Ohr is required for optimal stem nodulation….????

Author reply: Corrected.

  1. Line 301. How many nodulation assays have been done to obtain the results of fig 6B and C. And in Fig S1 and Fig. S3?

Author reply: Added.

  1. Lines 18, 49, 57, 294, 303, 343, 347, 353 Sesbania rostrata

Author reply: Corrected.

Reviewer 2 Report

Brief summary:

In this manuscript, authors identified and characterised the organic hydroperoxide resistance gene ohr (AZC_2977) and its negative regulator ohrR (AZC_3555) in A. caulinodans ORS571. They determinate that ohr is involved in the resistant to organic hydroperoxides, whereas ohrR is a negative regulator that in the presence of these compounds dimerizes and releases the promoter region of the ohr gene, which trigger gene expression. Authors also demonstrated that one of the two conserved cysteine residues of OhrR, Cys11, is critical for this behaviour. Finally, nodulation assays revealed that Ohr is required for a successful stem nodulation of A. caulinodans ORS571 with Sesbania rostrate. I think that English style is not bad but could be improved (I´m not an English native speaker). However, the manuscript must be amended, since some experimental procedures, references and statistical analyses are not displayed, and some feasible experiments could be performed in order to strengthen data showed in the manuscript. Besides, some statements in the introduction section are not correct. In my opinion, this paper is interesting and useful, but their relevance is partially obscured by several points that should be solved. For this reason, I suggest reconsideration after major revision.

Major points:

  • Introduction:
    • Lines 45-46: “ROS signals trigger the production of Nod factors, with the subsequent development of nodules and nitrogen fixation activity”. This sentence is disturbing and incorrect. In the paper of Shaw and Long (2003) is stablished that specific Nod Factors inhibits the production of ROS in a host legume. The production of Nod factors in rhizobia is induced by specific flavonoids released by compatible legume root. Please, reformulate this sentence accordingly.
  • Material and methods: some methods are not adequately described, making impossible to reproduce the experimental procedure:
    • References of bacterial culture media (lines 96, 97 and so on), of bacterial strains and plasmids (lines 104, 108, table S1) are missing.
    • A paragraph indicating how authors check mutant strains and how they perform bi or triparental conjugation is also missing.
    • Root nodulation protocols are not included in the section.
    • Section 2.9. is really confusing. Did you construct mutants or in trans complemented strains? Moreover, I think “mutant proteins” is not a correct term. Instead, use “mutated version of the proteins”. Please clarify through the manuscript.
  • Results:
    • A disk diffusion assay adding H2O2 and superoxide ion would delimitate the role of ohr in the resistant to OHPs in caulinodans ORS571.
    • It has been described that OhrR represses not only ohr but also its own ohrR expression and the expression of both genes is specifically induced by organic peroxides (BMC Microbiol 11, 100 (2011). doi.org/10.1186/1471-2180-11-100). Did you carry out any kind of expression assay in the presence of CuOOH in the ohrR mutant background to stablish same kind of regulation despite both ohr and ohrR are no located together?
    • Why did you not perform nitrogen fixation experiments in roots?

Minor points:

  • Introduction:
    • A sentence about autoregulation of nodulation (AON) mediated by ROS in compatible rhizobium-legume symbiosis should be included in the introduction section.
    • Is the same “lipid hydroperoxides” (line 58) than “organic hydroperoxides” (line 59)? Is the same “Cys-based, lipoyl-dependent” (line 67) than “thiol-dependent” (line 71)? Please, indicate and unify if that is the case.
    • “organic peroxide-inducible transcription repressor” is confusing.
  • Material and methods:
    • Please, indicate country and state for companies.
    • Line 100, indicate the spectrophotometer.
    • Section 2.5: indicate the program used for qPCR experiments.
    • Line 114: is CuOOH a kind of organic peroxide? Please, indicate.
    • Line 173: how do you measure acetylene and ethylene concentrations? Please, explain briefly including information of the device used for these experiments.
    • Line 153: final concentration is 100 nM, right?
    • Section 2.8: indicate how do you add DTT an CuOOH in some cases.
  • Results:
    • Section 3.1: gene IDs are missing.
    • Statistical analysis is missing in figure 2c.
    • Size bars are missing in figure 1A.
    • Section 3.1: indicate the positions of the conserved cys of the OhrR from caulinodans ORS571.
    • Lines 256-257: please, include these controls as supplementary figures.
    • Figure 3: only two statistical significances are displayed according to data. So, “∗ : Student t-test P < 0.05; ∗∗ : P < 0.01; ∗∗∗ : P < 0.005” should be replace by “∗ : Student t-test P < 0.05; ∗∗ : P < 0.005”.
  • Other sections:
    • In my opinion, figure S2 is relevant and should be in the main body of the manuscript.
    • Line 1 and so on: Ohr has no capital letter at the end indicative of gene name?
    • Line 4: include the strain name in the title.
    • Line 41: Rhizobium in italics is just a specific genus. Replace it for rhizobium or rhizobia.
    • Line 239: gene names are in italics.
    • Line 340. Which species?
    • In Sinorhizobium meliloti the inactivation of ohr did not affect symbiosis and nitrogen fixation (BMC Microbiol 11, 100 (2011). doi.org/10.1186/1471-2180-11-100). This finding strengthens results obtained in this manuscript, so include a sentence in the discussion section about this.

Author Response

Reviewer #2:

Brief summary:

In this manuscript, authors identified and characterised the organic hydroperoxide resistance gene ohr (AZC_2977) and its negative regulator ohrR (AZC_3555) in A. caulinodans ORS571. They determinate that ohr is involved in the resistant to organic hydroperoxides, whereas ohrR is a negative regulator that in the presence of these compounds dimerizes and releases the promoter region of the ohr gene, which trigger gene expression. Authors also demonstrated that one of the two conserved cysteine residues of OhrR, Cys11, is critical for this behaviour. Finally, nodulation assays revealed that Ohr is required for a successful stem nodulation of A. caulinodans ORS571 with Sesbania rostrate. I think that English style is not bad but could be improved (I´m not an English native speaker). However, the manuscript must be amended, since some experimental procedures, references and statistical analyses are not displayed, and some feasible experiments could be performed in order to strengthen data showed in the manuscript. Besides, some statements in the introduction section are not correct. In my opinion, this paper is interesting and useful, but their relevance is partially obscured by several points that should be solved. For this reason, I suggest reconsideration after major revision.

Author reply: Thanks for your suggestion. The paper was polished by Editage English language editing services.

Major points:

Introduction:

  1. Lines 45-46: “ROS signals trigger the production of Nod factors, with the subsequent development of nodules and nitrogen fixation activity”. This sentence is disturbing and incorrect. In the paper of Shaw and Long (2003) is stablished that specific Nod Factors inhibits the production of ROS in a host legume. The production of Nod factors in rhizobia is induced by specific flavonoids released by compatible legume root. Please, reformulate this sentence accordingly.

Author reply: Thanks for your suggestion, this sentence has been changed.

Material and methods:

  1. Some methods are not adequately described, making impossible to reproduce the experimental procedure: References of bacterial culture media (lines 96, 97 and so on), of bacterial strains and plasmids (lines 104, 108, table S1) are missing.

Author reply: Corrected.

  1. A paragraph indicating how authors check mutant strains and how they perform bi or triparental conjugation is also missing.

Author reply: Added.

  1. Root nodulation protocols are not included in the section.

Author reply: Added.

  1. Section 2.9. is really confusing. Did you construct mutants or in trans complemented strains? Moreover, I think “mutant proteins” is not a correct term. Instead, use “mutated version of the proteins”. Please clarify through the manuscript.

Author reply: Thank you for your suggestion. Added

Results:

  1. A disk diffusion assay adding H2O2 and superoxide ion would delimitate the role of ohr in the resistant to OHPs in caulinodans ORS571.

Author reply: Thank you for your suggestion. We have already tested the sensitivity of A. caulinodans strains in response to H2O2, and there was no differece between WT and Δohr strain. This result agrees with Ohr is specific sensitive to organic peroxide. These data are not shown in this manuscript.

  1. It has been described that OhrR represses not only ohr but also its own ohrR expression and the expression of both genes is specifically induced by organic peroxides (BMC Microbiol 11, 100 (2011). doi.org/10.1186/1471-2180-11-100). Did you carry out any kind of expression assay in the presence of CuOOH in the ohrR mutant background to stablish same kind of regulation despite both ohr and ohrR are no located together?

Author reply: Thank you for the suggestion. We performed RNA-seq of A. caulinodans WT and ΔohrR in response to oxidants (data not shown). The results showed that OhrR represses its own expression slightly (less than 2 fold). We added some discussion in the second paragraph of Discussion.

  1. Why did you not perform nitrogen fixation experiments in roots?

Author reply: We focus on stem nodules, it is more likely to be attacked by ROS. The number of root nodules and pink nodules were no difference. It seems ohr did not affect root nodule formation then we did not test nitrogen fixation ability of root nodule. Sorry!

Minor points:

Introduction:

  1. A sentence about autoregulation of nodulation (AON) mediated by ROS in compatible rhizobium-legume symbiosis should be included in the introduction section.

Author reply: Added.

  1. Is the same “lipid hydroperoxides” (line 58) than “organic hydroperoxides” (line 59)? Is the same “Cys-based, lipoyl-dependent” (line 67) than “thiol-dependent” (line 71)? Please, indicate and unify if that is the case.

Author reply: Corrected.

  1. “organic peroxide-inducible transcription repressor” is confusing.

Author reply: Changed.

Material and methods:

  1. Please, indicate country and state for companies.

Author reply: Added.

  1. Line 100, indicate the spectrophotometer.

Author reply: Added.

  1. Section 2.5: indicate the program used for qPCR

Author reply: Added.

  1. Line 114: is CuOOH a kind of organic peroxide? Please, indicate.

Corrected.

  1. Line 173: How do you measure acetylene and ethylene concentrations? Please, explain briefly including information of the device used for these experiments.

Author reply: Added.

  1. Line 153: final concentration is 100 nM, right?

Author reply: 20 nM. Described in text.

  1. Section 2.8: indicate how do you add DTT and CuOOH in some cases.

Author reply: Added.

Results:

  1. Section 3.1: gene IDs are missing.

Author reply: Added

  1. Statistical analysis is missing in figure 2c.

Author reply: Added.

  1. Size bars are missing in figure 1A.

Author reply: Added.

  1. Section 3.1: indicate the positions of the conserved cys of the OhrR from caulinodans ORS571.

Author reply: Added.

  1. Lines 256-257: please, include these controls as supplementary figures.

Author reply: Controls are shown in Figure 4B, lane 7 and lane 8.

  1. Figure 3: only two statistical significances are displayed according to data. So, “∗ : Student t-test P < 0.05; ∗∗ : P < 0.01; ∗∗∗ : P < 0.005” should be replace by “∗ : Student t-test P < 0.05; ∗∗ : P < 0.005”.

Author reply: Corrected

Other sections:

  1. In my opinion, figure S2 is relevant and should be in the main body of the manuscript.

Author reply: Figure S2 was transferred to Figure 5B in the main body

  1. Line 1 and so on: Ohr has no capital letter at the end indicative of gene name?

Author reply: “Ohr” has no capital letter at the end

  1. Line 4: include the strain name in the title.

Author reply: Changed.

  1. Line 41: Rhizobium in italics is just a specific genus. Replace it for rhizobium or rhizobia.

Check and corrected

  1. Line 239: gene names are in italics.

Corrected

  1. Line 340. Which species?

Author reply: Species were indicated

  1. In Sinorhizobium meliloti the inactivation of ohr did not affect symbiosis and nitrogen fixation (BMC Microbiol 11, 100 (2011). doi.org/10.1186/1471-2180-11-100). This finding strengthens results obtained in this manuscript, so include a sentence in the discussion section about this.

Author reply: Thanks for your suggestion, a sentence was added in the discussion.

Reviewer 3 Report

Overall very nice research. The theme seems to be fully disclosed by the authors. However, I see some minor details that can be improved:

Introduction:

  1. Lines 41-47. Spelling Rhizobium confuses with the particular genus of bacteria, but I guess that authors meant rhizobia in general.
  2. Lines 87-91. I would like to see more explanation on relevance of the research.

Materials and Methods

  1. The paragraph about “bioinformatics analyses” from 3.1 should be added.
  2. Information about replicates should be included in materials too, not only in results.
  3. 5, line 130. Is it appropriate to use multicopy 16S in quantitative pcr analysis?
  4. 9, line 165. Should spell “plasmids”, not plasimds
  5. 10. Information about seed/plant replicates is missing in both materials and results.

Results.

  1. Figure 1. I don’t see a necessity of panel A. Instead, schemes of constructed plasmids and recombination process would be a nice addition helping to understand the cloning process.
  2. 6, line 292-293. Sentence “Approximately 108 cells …” should be transferred to 2.10 in materials
  3. Figure 6. No mention of experiment replicates

Discussion nicely summarizes all the experiment data, but phrase “Our work improves the knowledge” in line 353 depreciates the huge experimental work done. Is there more substantial explanation of the importance of research?

Author Response

Reviewer #3:

Overall very nice research. The theme seems to be fully disclosed by the authors. However, I see some minor details that can be improved:

Introduction:

  1. Lines 41-47. Spelling Rhizobium confuses with the particular genus of bacteria, but I guess that authors meant rhizobia in general.

Changed to “rhizobia”

  1. Lines 87-91. I would like to see more explanation on relevance of the research.

Author reply: More explanations were added in the introduction.

Materials and Methods

  1. The paragraph about “bioinformatics analyses” from 3.1 should be added.

Author reply: Thanks for your suggestion. Added.

  1. Information about replicates should be included in materials too, not only in results.

Author reply: Replicates were indicated in the main text

  1. 5, line 130. Is it appropriate to use multicopy 16S in quantitative pcr analysis?

Author reply: The 16s rRNA gene is stable for normalizing mRNA in qRT-PCR according to previous study of A. caulinodans.

(Ling, Jun, et al. "Plant nodulation inducers enhance horizontal gene transfer of Azorhizobium caulinodans symbiosis island." Proceedings of the National Academy of Sciences 113.48 (2016): 13875-13880.)

  1. 9, line 165. Should spell “plasmids”, not plasimds

Corrected

  1. Information about seed/plant replicates is missing in both materials and results.

Author reply: Add description in the main text

Results.

  1. Figure 1. I don’t see a necessity of panel A. Instead, schemes of constructed plasmids and recombination process would be a nice addition helping to understand the cloning process.

Author reply: Thanks for your suggestion. panel A has changed. The scheme of constructed plasmids and recombination added in supplemental material (Figure S1).

  1. 6, line 292-293. Sentence “Approximately 108 cells …” should be transferred to 2.10 in materials

Author reply: Corrected

  1. Figure 6. No mention of experiment replicates

Author reply: Added

  1. Discussion nicely summarizes all the experiment data, but phrase “Our work improves the knowledge” in line 353 depreciates the huge experimental work done. Is there more substantial explanation of the importance of research?

Author reply: Thanks for your suggestion, the final paragraph has changed.

Round 2

Reviewer 2 Report

This paper is an amended version of the work sent previously to Genes. In this new manuscript the instructions, suggestions and corrections that I indicated in the first evaluation have been almost completely addressed (or at least well argued). Thus, I recommend accept the paper for publication with few modifications detailed below.

Line 41: “Gram-negative bacterium” should be replace by “Gram-negative bacteria”

Lines 45-46: “ROS mediate Nod Factor responses” What this sentence means?

Lines 201: N2-fixing.

Root nodulation protocols remain not included in the material and method section, right? Please, add it.

3.1. Section: Ohr and OhrR gene IDs of Xanthomonas campestris, Pseudomonas aeruginosa, Agrobacterium tumefaciens, and Sinorhizobium meliloti remain missing. Please, add them.

Figure 2c: Did you find statistical differences among CuOOH treatments in all concentrations or just in the last one?

Line 398: “…and nitrogen fixation”. Add reference.

Table S1: I think is not correct to indicate as source “Lab collection” when these strains are well characterized worldwide. Please, reference.

Author Response

Line 41: “Gram-negative bacterium” should be replace by “Gram-negative bacteria” Author reply: Corrected Lines 45-46: “ROS mediate Nod Factor responses” What this sentence means? Author reply: Corrected. Lines 201: N2-fixing. Author reply: Corrected. Root nodulation protocols remain not included in the material and method section, right? Please, add it. Author reply: added. 3.1. Section: Ohr and OhrR gene IDs of Xanthomonas campestris, Pseudomonas aeruginosa, Agrobacterium tumefaciens, and Sinorhizobium meliloti remain missing. Please, add them. Author reply: Added. Figure 2c: Did you find statistical differences among CuOOH treatments in all concentrations or just in the last one? Author reply: In all concentrations, but just show in the highest concentration. Line 398: “…and nitrogen fixation”. Add reference. Author reply: Added. Table S1: I think is not correct to indicate as source “Lab collection” when these strains are well characterized worldwide. Please, reference. Author reply: Corrected.
